# CricShotClassify: An Approach to Classifying Batting Shots from Cricket Videos Using a Convolutional Neural Network and Gated Recurrent Unit

**DOI:** 10.3390/s21082846

**Published:** 2021-04-18

**Authors:** Anik Sen, Kaushik Deb, Pranab Kumar Dhar, Takeshi Koshiba

**Affiliations:** 1Department of Computer Science & Engineering, Chittagong University of Engineering & Technology (CUET), Chattogram 4349, Bangladesh; aniksen.cuet09@gmail.com (A.S.); pranabdhar81@cuet.ac.bd (P.K.D.); 2Department of Computer Science & Engineering, Premier University, Chattogram 4000, Bangladesh; 3Faculty of Education and Integrated Arts and Sciences, Waseda University, 1-6-1 Nishiwaseda, Shinjuku-ku, Tokyo 169-8050, Japan; tkoshiba@waseda.jp

**Keywords:** batting shots, deep learning, transfer learning, convolutional neural network, gated recurrent unit

## Abstract

Recognizing the sport of cricket on the basis of different batting shots can be a significant part of context-based advertisement to users watching cricket, generating sensor-based commentary systems and coaching assistants. Due to the similarity between different batting shots, manual feature extraction from video frames is tedious. This paper proposes a hybrid deep-neural-network architecture for classifying 10 different cricket batting shots from offline videos. We composed a novel dataset, CricShot10, comprising uneven lengths of batting shots and unpredictable illumination conditions. Impelled by the enormous success of deep-learning models, we utilized a convolutional neural network (CNN) for automatic feature extraction, and a gated recurrent unit (GRU) to deal with long temporal dependency. Initially, conventional CNN and dilated CNN-based architectures were developed. Following that, different transfer-learning models were investigated—namely, VGG16, InceptionV3, Xception, and DenseNet169—which freeze all the layers. Experiment results demonstrated that the VGG16–GRU model outperformed the other models by attaining 86% accuracy. We further explored VGG16 and two models were developed, one by freezing all but the final 4 VGG16 layers, and another by freezing all but the final 8 VGG16 layers. On our CricShot10 dataset, these two models were 93% accurate. These results verify the effectiveness of our proposed architecture compared with other methods in terms of accuracy.

## 1. Introduction

Videos uploaded in large volumes onto various streaming platforms such as YouTube and Vimeo play a critical role in the era of data communication. Simultaneously, analyzing data from these videos is a critical issue. Making important decisions while constantly monitoring what is changing in their frames is a primary concern when working with video data. The same can be said for various sports videos. Due to the various types of camera positions and movements, there are numerous opportunities to experiment with sports videos. This data analysis is often applied to sports-related work, which is an emerging issue with multimedia amplification. Several attempts were made to classify different actions from sports [1,2,3,4]. Cricket is one of those, thriving with a huge following all over the world. A tournament such as the ICC Men’s Cricket World Cup 2019 hits a global average audience of no less than 1.6 billion, making it one of the most viewed events ever [5]. This vast popularity and enormous fan base create a commercial aspect for researching cricket data. Broadcasters can make critical decisions by scrutinizing this massive volume of frequently uploaded videos. To automatically find cricket viewers, for example, it is necessary to analyze the video content. Among all regions, cricket is one of the most crowd-pleasing sports in Asia. As a consequence, online viewers and spectators are considered to be significant customers for broadcasters. Although it is significant to examine cricket from a commercial standpoint, little has been done in this regard so far.

Cricket videos require analysis to develop an unbiased, equitable, and sensor-based commentary system. Moreover, detecting cricket on the basis of different types of cricket shots can be helpful for both coaches and sports analysts. Coaches and cricket experts have to regularly understand the weaknesses of different teams and adjust their game plans on the basis of those findings. In this context, it is vital to extract information from cricket data. Considering all the mentioned aspects, we chose cricket as the base sport and considered 10 different batting shots: cover drive, defensive shot, flick shot, hook, late cut, lofted shot, pull, square cut, straight drive, and sweep. The similarity of frames between different videos makes it challenging to accurately categorize batting shots. The leading contributions of our work are summarized below:We developed the novel dataset CricShot10 consisting of 10 different cricket shots. To the best of our knowledge, this is the highest number of distinct cricket shots in a single work.We designed a custom convolutional neural network (CNN)–recurrent neural network (RNN) architecture to classify the 10 different cricket shots.We investigated the impact of different types of convolution on our dataset.We investigated various transfer learning models to accurately classify batting shots.

The rest of this paper is ordered as follows. Section 2 discusses related work. Section 3 describes the working flow of developing a deep-neural-network architecture for cricket batting-shot categorization (CBSC). Section 4 outlines the experiment results and discussion, including the dataset-generation process and the evaluation of various proposed model architectures. Lastly, Section 5 concludes the paper by discussing future work.

## 2. Related Work

Sports-based research is popular nowadays due to its commercial benefits and large audience. Several methods have been developed in recent years to classify different sports actions from videos. Some attempts were recently made to extract information from cricket sports videos, some of which were based on categorizing different cricket shots.

In [6], eight different cricket shots were classified using deep convolutional neural networks. A 2D CNN, 3D CNN, and long short-term memory (LSTM) recurrent neural network (RNN) were used to categorize different cricket shots. Transfer learning from a pretrained VGG model was also used. The highest accuracy of 90% was achieved using a dataset of 800 cricket shots. However, the best model had a low accuracy of 85% for the hook shot. In [7], a motion-estimation approach was proposed to classify cricket shots. Eight classes of angle ranges were defined to detect cricket shots. Accuracy was comparatively much lower: the highest accuracy of 63.57% was achieved for the off drive, and 53.32% accuracy was achieved for the hook. In [8], wearable technology was introduced to classify various cricket shots. The fundamental objective of that work was to make a quality-assessment system of various cricket shots. Hierarchical representation was used on the basis of specific aspects of cricket shots such as foot position and shot direction. Five levels of grouping actions were classified using decision tree (DT), support vector machine (SVM), and k-nearest-neighbor (k-NN) algorithms. An 88.30% class-weighted F1 score was attained using the best-performing classifiers per level. By ignoring the assessment using the real match videos, this work only considers videos collected from a training session. Foysal et al. [9] proposed a CNN model to classify six different cricket shots. They experimented on their custom-developed dataset. Semwal et al. [10] proposed a model to detect batting shots: straight drive, cover drive, and lofted on drive for both left- and right-handed batsmen. A pretrained CNN and multiclass SVM were utilized. Their proposed model attained 83.098% accuracy for 3 categories of right-handed shots and 65.186% accuracy for 3 categories of left-handed shots. The proposed model misclassified a few shots such as left cover drive, left straight drive, and right straight drive due to the limited dataset.

Quite a few methods were attempted to incorporate different events in cricket videos [11,12]. Harikrishna et al. [11] proposed an approach that initially segmented a cricket video into shots and then used an SVM to classify the detected shot into 4 events, namely, run, four, six, out. With the proposed approach, recorded accuracy was 87.8%. Their proposed model had inferior accuracy for the six events. Kolekar et al. [12] proposed a hierarchical framework for detecting cricket events: excitement detection, replay detection, field-view detection, field-view classification, close-up detection, close-up classification, and crowd classification. The authors mentioned misclassification near the shot boundaries. Premaratne et al. [13] proposed a structural approach for identifying various cricket events from a full-length cricket match. For detecting events, k-nearest-neighbor, sequential-minimal-optimization, decision-tree, and naive Bayes classifiers were used. However, the authors mentioned human interpretation for removing replay frames and highlights between delivery.

Some studies were conducted for automatic highlight extraction from cricket videos [14,15,16,17,18,19]. In [14], video and audio features were combined. The authors first used rule-based induction to detect exciting audio clips. Afterwards, a video summarization step was performed using the decision tree. Their proposed method attained 90% average accuracy. In [15], the authors proposed a hierarchical method for automatic highlight generation from cricket videos. Association-rule mining was used to extract semantic concepts. The authors mentioned some important features such as the umpire’s gesture detection and camera motion estimation as future work. Video-shot-detection, sound-detection, and scoreboard-recognition-based cricket-game summation methods were proposed in [16]. In [17], a novel approach was proposed that could detect highlights in sports videos. A color histogram (CH) and a histogram of oriented gradients (HOG) were used in their work. A novel summarization scheme was proposed in [18]. In this work, boundary and wicket events were extracted on the basis of bowling and score change. Ads and replay events were removed by calculating the temporal derivative. A simple, real-time, highlight-extraction, hierarchical approach was proposed in [19]. In this work, a combination of motion and color features was used within a multispatial framework. Ringis et al. [20] proposed a method that automatically detected bowler run-up sequences (BRSs) using the oriented fast and rotated brief (ORB) method in order to generate cricket highlights. There are some limitations to their proposed method: training is required at the beginning of a match, there is only detection of players as key points, there is no use of orientation parameters, and there is only training and detecting of BRS views. Rafiq et al. [21] proposed a transfer-learning-based method that classified sports video scenes to generate video summarization. Over a smaller dataset containing only cricket scenes, the proposed Alexnet CNN-based approach attained 99.26% accuracy.

So far, only a few experiments involving sports other than cricket have been conducted. Steels et al. [22] proposed a method that uses a convolutional neural network and accelerometer data to recognize 9 different badminton actions. The sensor placement in the wrist and upper arm had a weighted accuracy of 93% and 96%, respectively, which was marginally lower than the average accuracy of 98%. Rangasamy et al. [23] proposed a model using the VGG16 pretrained network to classify 4 hockey activities, namely, free hit, goal, penalty corner, and long corner. The highest accuracy of 98% was reached after running the model for 300 epochs. The proposed model was slightly confused about the difference between a free hit and a long corner because these activities have a visually similar pattern in terms of the player’s position and appearance. Junjun et al. [24] proposed a method for recognizing 3 types of basketball actions: shoot, pass, and catch. Using an adaptive multilevel classification technique, the proposed model attained 92.3% accuracy. However, this study only considered a few exploratory examples. Gu et al. [25] proposed a method for fine-grained video action recognition from basketball games. The research looked at 3 broad actions—dribbling, passing, and shooting—which are further divided into 26 fine-grained actions. However, in a broad category, the average precision of subclasses significantly differs due to their specific characteristics.

## 3. Workflow of Proposed Architecture

This section describes the construction process of the cricket batting-shot categorization (CBSC) model. For all our conducted experiments, 15 randomly sampled frames from each video were considered. All inputs were preprocessed to have pixel values ranging 0–1. Initially, we built our CNN–RNN network from scratch and, after many trials, came up with two different architectures. The first was based on a conventional CNN and gated recurrent unit (GRU) as the recurrent network. The second was built using a CNN variant, dilated CNN (DCNN), and GRU. Afterwards, we examined transfer learning from several pretrained models to investigate the best model to attain higher accuracy. The transfer learning process was initiated by freezing all layers of the different pretrained models. We considered the pretrained models VGG16, InceptionV3, Xception, and DenseNet169 for our work. Experiment results showed that the VGG-based model had comparatively better accuracy than that of the others. Hence, we performed an in-depth exploration considering VGG16-based CNN–RNN models. The overall process is depicted in Figure 1.

### 3.1. Custom CNN–GRU Architecture

Our first trial was to build a custom architecture from scratch. We considered frames of 180×180 px and 15 sequences per video to build our custom model. At present, many accurate diagnoses are being produced in the realm of computer vision. The revolution of deep-learning-based approaches is the underlying reason for this achievement. For many years, researchers have been looking for a way to reduce hand-engineered feature-extraction approaches. CNNs are now pushing all boundaries of this problem with tremendous accuracy in different computer-vision tasks. Dominant spatial features were extracted using CNN. Four steps, namely, convolution, pooling, flattening, and a fully connected layer are generally required to perform feature extraction from images.

The input image is convolved with some filter and rolled over the whole image in the convolution step. For example, if a 4×5 sized image is convolved by a 2×2 filter, it produces an output of a 3×4-sized image, as shown in Figure 2. We can control the movement of filters across the whole image by specifying a value for stride. Padding values can be set to mitigate the disadvantages of shrinking outputs and losing information. Pooling is an operation that downsamples its given input. By a flattening operation, features are converted to a linear vector. This flattened vector is then passed to a fully connected layer and a prediction is made for an image-classification task.

We used a 3×3 filter size for our CNN, and a 180×180 px sized frame was considered throughout all experiments. The rectified linear unit (ReLU) activation function, defined in Equation (Equation 1), was used in the CNN layer.
(1)$$f\left(x\right)=\left\{\begin{array}{cc} 0 & x\leq 0\\ x & x>0. \end{array}\right.$$

We used 2×2 max pooling to downsample the inputs. Inputs were padded to have the same dimensions after each convolutional operation to prevent the model from losing spatial information. Batch normalization was used after each convolutional block to reduce the number of training steps [26]. The proposed CNN architecture is presented in Table 1.

Our CNN model’s output was then fed into a time-distributed wrapper. Weights were updated for all frames if frames from a video were individually passed to their adjacent layer. As a result, differences between video frames were meaningless. Consequently, a time-distributed layer was introduced that provided the option to share layers across video frames. The time-distributed layer’s input shape should be in the form of a 5D tensor. Following that, the time-distributed flatten operation generated an output shape that could be passed to a recurrent neural network (RNN). The time step value in our case was 15, and 12,000 features were passed in each time step.

To detect actions from a video, the coordinate feature alone is insufficient. It is necessary to know what happened in the previous frame to correctly classify action in a video. Various RNNs were proposed for maintaining this temporal dependency, the input of which is a 3D vector of format (samples, time steps, units).

Traditional RNNs, on the other hand, suffer from the problems of vanishing gradient and exploding gradient [27]. As a result, the LSTM [28] was proposed. The LSTM algorithm performed admirably in tracking the vanishing gradient and exploding gradient problems, as well as long-term events. Through its various gate concept, LSTM solves the problem of time dependency. Following that, researchers created GRU [29], a simplified version of LSTM. GRU demonstrated unrivaled success in long-term feature dependency using fewer gates than LSTM. It also reduces the number of trainable parameters by using fewer gates, simplifying the network. The update, reset, and current-memory gates were used in our study. The update gate helps the model determine how much previous knowledge should be passed on to future time steps. Furthermore, the model learns how much of the new state is simply a copy of the old state value, removing the risk of a vanishing gradient. The reset gate aids the model in determining how much previous knowledge it should forget. The current-memory gate calculates the current state value by combining the previous hidden state with the current input. Figure 3 illustrates a typical GRU cell diagram.

The equations for the update gate, reset gate, current memory state, and final memory are given in Equations (Equation 2)–(Equation 5), respectively:(2)$$z_{t}=\sigma \left(W_{z}\cdot \left[h_{t-1},x_{t}\right]+b_{z}\right)$$
(3)$$r_{t}=\sigma \left(W_{r}\cdot \left[h_{t-1},x_{t}\right]+b_{r}\right)$$
(4)$$\tilde{h_{t}}=\tanh\left(W\cdot \left[r_{t}*h_{t-1},x_{t}\right]+b\right)$$
(5)$$h_{t}=\left(1-z_{t}\right)*h_{t-1}+z_{t}*\tilde{h_{t}},$$
where $W_{z}$, $W_{r}$, and *W* represent weight matrices; $h_{t-1}$ is the hidden state of the previous time step; $x_{t}$ is the current input vector; σ and tanh are the sigmoid and tanh activation function; ∗ represents the Hadamard product; and $b_{z}$, $b_{r}$, and *b* are biases.

The architecture of the CNN–GRU model is given in Table 2. In the final layer, a softmax activation function, defined by Equation (Equation 6), was used that returns the probability distribution of each batting shots that sums up to 1.
(6)$$\sigma {\left(z\right)}_{j}=\frac{e^{z_{j}}}{\sum_{k=1}^{k}e^{z_{k}}}\;\mathrm{for}\;j=1,\ldots ,k$$

For all experiments, stochastic gradient descent (SGD) optimizer with 0.0001 learning rate and 0.9 momentum was used. A categorical cross-entropy loss function was used, which is defined in Equation (Equation 7).
(7)$$L\left(\hat{y_{i}},y\right)=-\sum_{i=1}^{K}y_{i}\log\hat{\left(y_{i}\right)},$$
where *K* is the total number of classes, $y_{i}$ is the desired one-hot encoded vector, and $\hat{y_{i}}$ is the predicted model output vector. During backpropagation, the gradient starts to backpropagate through the derivative of the loss function with respect to the output of the softmax layer. The merged CNN–GRU architecture is outlined in Table 2.

### 3.2. Custom DCNN–GRU Architecture

Dilated convolution is a variation of convolution [30] that uses a defined gap. For a convolutional operation to be termed as dilated convolution, the gap factor must be greater than or equal to 2. If the dilation factor is 1, then it is termed as a traditional convolutional operation. Sports videos of different batting shots contain objects other than only a batsman’s action. After hitting a shot, the ball’s direction can be an important aspect of defining the cricket shot. Thus, to have a larger overview of the scene content, dilated convolution was used here. A dilation rate of 2 was used to build our custom architecture. The valid padding method was used to incorporate dilation and reduce the spatial dimension after each convolutional layer. A max-pooling operation was carried out using a size of 2×2. Batch normalization was utilized after each convolutional block. The ReLU activation function was applied to introduce nonlinearity after each convolutional layer. Dilation operations with factors of 1, 2, and 4 are depicted in Figure 4a–c, respectively.

Our proposed DCNN model architecture is summarized in Table 3.

Afterwards, the outputs from the DCNN network and flattened layer were enclosed in a time-distributed wrapper. A GRU layer with 180 hidden units was used here. The combined DCNN–GRU architecture is highlighted in Table 4.

### 3.3. Transfer-Learning-Based Proposed Architecture

The term “transfer learning” applies to situations in which knowledge learned from one problem can be applied to resolve another problem. In the past few years, pretrained models saw high acceptance in computer-vision tasks such as real-time object recognition and image classification. As a result of learning on a large number of datasets such as ImageNet [31], several pretrained models are conducive to solving similar problems. High accuracy is not available using shallow learning models because they require a large amount of data. Handling this enormous dataset is almost impossible due to a shortage of available data resources. Since most pretrained models are generated from a massive dataset containing a few million samples, this typical use of the developed models results in a baseline for further data analysis. Layer freezing is applied to disable the weight updating of the layers of models. Furthermore, the weights of specific layers can be kept frozen, allowing for weights based on the custom dataset to accordingly change. Following initialization of the pretrained models, other layers can also be integrated. In our work, we used transfer learning from pretrained VGG16 [32], InceptionV3 [33], Xception [34], and DenseNet [35] models. These models were developed on the basis of various concepts, e.g., VGG16 was designed using a small filter of 3×3; InceptionV3 was focused on reduced trainable parameters; Xception was developed on the basis of depthwise separable convolution; and DenseNet was based on connecting layers in a feed-forward fashion. We disallowed weight updates of all pretrained models by freezing all layers.

ImageNet dataset weights were considered for all pretrained models, and all fully connected layers were discarded. A flattening layer was introduced after loading a pretrained model, which converted the input shape into a 1D array. The time-distributed wrapper was then initiated to wrap the pretrained model layers and the flattening layer, which allowed for sharing the layer weights for all sampled video frames. Afterwards, the GRU layer was appended to trace temporal changes across video frames. Furthermore, 3 consecutive dense layers with hidden units of 512, 128, and 64 were attached to the model, and ReLU was used as the activation function that accelerated the model capacity by introducing nonlinearity. Lastly, a prediction was obtained using the softmax classifier that assigned a probability value to each batting shot that summed to 1.

Figure 5 depicts the generalized model architecture for combined transfer learning models and GRU.

### 3.4. Proposed VGG-Based Architecture

VGG is a pretrained CNN architecture that uses small kernel of 3×3. The originally proposed VGG16 architecture used 224×224 input images. After 5 convolutional blocks, 3 fully connected layers were attached; first, 2 fully connected layers had 4096 hidden units, while the third had 1000 layers, representing the number of output classes. The originally proposed VGG16 architecture is depicted in Figure 6. Comparing other transfer learning models—InceptionV3, Xception, and DenseNet169—using VGG16, a validation accuracy of 86% could be achieved, which motivated us to further explore VGG16 models. For our case, we considered a 180×180 px input image, and all fully connected layers were discarded. Thus, the output from the final layer became (5, 5, 512). This architecture is represented in Figure 7.

A CNN’s initial layers extract low-level features such as colors and lines. The deeper layers detect distinguishable features that are more incomprehensible. In Figure 8, sequential frames for each batting shot is depicted. Several low–high-level feature maps for different batting shots generated by the VGG16 model are shown in Figure 9. Unlike the approach described in Section 3.3 where all layers were frozen, we allowed some of the final layers to update weights. Then, 2 models were derived: Model 1 was designed by making the final 4 layers of VGG16 trainable, and Model 2 by making the final 8 layers of VGG16 trainable. All remaining configurations of the GRU layer, fully connected layers, were kept in a similar fashion as that highlighted in Figure 5. The two newly executed models are summarized in Table 5.

## 4. Result and Discussion

In this section, we discuss the hardware configuration, dataset generation, and various accuracy measures.

### 4.1. Hardware Configuration

The overall experiment was conducted on an AMD Ryzen 7 2700X eight-core 3.7 GHz processor. The operating system was Ubuntu version 20.04, with 32 GB RAM and an Nvidia GeForce RTX 2060 Super with 8 GB of GPU memory. Keras with a TensorFlow back-end was used.

### 4.2. Dataset Generation

This work obtained 10 different cricket shots from different publicly accessible sources. We termed our dataset CricShot10. Only offline videos were taken into account for this project. Our novel dataset comes from a variety of ICC events (ODI, Test, and T20 matches), as well as the Indian Premier League (IPL), Bangladesh Premier League (BPL), and Big Bash League (BBL). Our dataset comprised sports played in various parts of the world and different types of video illumination. Batting shots from both left- and right-handed batsmen were taken into account. We considered variable instead of fixed video lengths to make it more applicable. Our dataset had a mean duration of 2.56 s and a standard deviation of 0.97 s. The entire dataset had a minimal duration of 1.0 s and a maximal duration of 7.72 s. Videos with a resolution of 1280×720 pixels were included in our dataset, and 35 videos of each batting shot constituted the test set. Separate matches of the different formats were collected to constitute the training set. Table 6 summarizes our custom-built dataset.

Sequential frames are shown in Figure 8.

### 4.3. Evaluation

This section describes various techniques used to evaluate our proposed models. We used precision, recall, and F1 score accuracy measures to compare our proposed architectures.

Precision is an accuracy measure of any machine-learning model that assesses the fraction of correctly classified samples among all samples classified as positives. The formula for precision is given in Equation (Equation 8).
(8)$$Precision=\frac{\mathrm{True}\;\mathrm{Positive}}{\mathrm{True}\;\mathrm{Positive}+\mathrm{False}\;\mathrm{Positive}}.$$

The recall of a model quantifies the number of correct positive predictions out of all positive predictions that could be made and is defined by Equation (Equation 9).
(9)$$Recall=\frac{\mathrm{True}\;\mathrm{Positive}}{\mathrm{True}\;\mathrm{Positive}+\mathrm{False}\;\mathrm{Negative}}.$$

The F score, also known as F1 score, is an accuracy measure that combines precision and recall and is designated as the harmonic mean of a model’s precision and recall. The formula for the F score is given by Equation (Equation 10).
(10)$$F-\mathrm{Score}=2\times \frac{\mathrm{Precision}\times \mathrm{Recall}}{\mathrm{Precision}\;+\;\mathrm{Recall}}.$$

### 4.4. Discussion

Results, including the parameters and performance of the various models, are highlighted in this section. The F1 score is referred to as accuracy because it combines precision and recall. First, we provide an overview of the proposed CNN–GRU and DCNN–GRU models. According to the experiment results shown in Table 7, the DCNN–GRU network, with its comparatively higher trainable parameters, achieved a slightly higher accuracy of 83% than that of the CNN–GRU model (82%).

We investigated various pretrained CNN models: VGG16, InceptionV3, Xception, and DenseNet169, and all layers were kept nontrainable. Combining GRU with the pretrained models, performance was assessed. Experiment results, as shown in Table 8, showed that the VGG16–GRU-based model had the best accuracy of 86%.

Lastly, two models were derived from VGG16 by allowing for the weight update of some intermediate layers. Model 1 was derived by allowing for a weight update for the final 4 layers, and Model 2 by allowing for a weight update for the final 8 layers. Although both models were 93% accurate, as recorded in Table 9, Model 1 consumed less of the trainable parameters.

The confusion matrix for the two derived models from VGG16 is represented in Figure 10a,b.

Although VGG16 models attained higher accuracy than that of other models, there were still some false predictions as every model learns differently because of various hyperparameters and training data. Interpreting the actual reason for this misprediction is not possible due to the neural network’s black-box nature. For instance, the confusion matrix portrayed in Figure 10a demonstrated that Model 1 classified 4 videos of flick shots as sweep shots. Despite trying to find a reason for this, we did not reach a definite conclusion. We can only assume that we could lessen the error by incorporating similar videos in the training set. Moreover, due to the extensive similarity between pull shot and hook shot, Model 2, as exhibited in Figure 10b, classified 4 videos of hook shots as pull shots.

Since the F1 score incorporates both precision and recall, we refer to F1 score as the accuracy measure. Our CNN–GRU model, which we created ourselves, had an accuracy of 82%, while the DCNN–GRU model had an accuracy of 83%. These two models had lower accuracy than that of transfer-learning models because the dataset was not broad enough for a deep-learning model to achieve high accuracy. By extending the dataset, accuracy can be further improved. Since pretrained models are built using enormous, diverse scene conditions, their weights must distinguish important features for other tasks. As a result of initializing the ImageNet weights unchanged, the VGG16 model achieved an accuracy of 86%. Initial layers extract low-level features such as color, texture, edges, and lines. In contrast, deeper layers extract distinguishable features. As a result, we left the initial weights untouched while enabling weight updates for some of the final layers. The network changed its weights on the basis of our samples, achieving the highest accuracy of 93%.

Accuracy-performance graphs for Model 1 and Model 2, as shown in Figure 11a,b, elucidate that, for some initial epoch, test accuracy was higher than training accuracy. After some epochs, training accuracy was higher than test accuracy. Thus, by training Model 1 for 65 epochs, it attained 92.79% test accuracy. Similarly, Model 2 attained a test accuracy of 92.87% after 79 epochs. After no improvements in test accuracy for 5 consecutive epochs, both models terminated the training.

When training the models, we used a batch size of 3. We used a stochastic gradient descent (SGD) optimizer for all of our proposed hybrid models with a learning rate of 0.0001 and momentum of 0.9. Various callback functions were used throughout our experiments. We trained all models several times, with each time loading the previous best model. The early stopping callback function stopped training the model after there was no improvement in validation accuracy for 5 consecutive epochs. Model checkpoint callback was used to save the best model, monitoring the validation accuracy after each epoch. The accuracy of our proposed combined CNN–GRU architecture and DCNN–GRU is depicted in Table 7. Experiment results show that our proposed CNN–GRU architecture achieves accuracy of 81%, whereas the DCNN–GRU architecture attained an accuracy of 83%. Therefore, dilated convolution achieved slightly higher accuracy than conventional convolution because it allowed for the network to have a more expansive view of the picture.

Afterwards, we implemented transfer learning from pretrained models VGG16, InceptionV3, Xception, and DenseNet169. We applied layer freezing to all the remaining convolutional layers after dropping all fully connected layers so that weights were not updated during backpropagation. We then added the GRU layer and fully connected layers. The configuration is shown in Figure 5. The accuracy of different transfer learning and GRU-based combination models is outlined in Table 8. By achieving 86% accuracy, the VGG16-based combination model directed us to penetrate further to examine if accuracy could be improved. We then developed two models from VGG16 and GRU by allowing some layers to update the weights. A CNN’s initial filters learn low-level features such as color, texture, and lines, and deeper filters learn descriptive features. Thus, we allowed the weight upgradation of some final layers and derived two models. Model 1 was executed by layer freezing except for the final 4 VGG16 layers, and Model 2 by layer freezing except for the final eight VGG16 layers. The arrangement of the GRU and fully connected layers was kept the same as that in the other models. The confusion matrix for Model 1 is given in Figure 10a, and the confusion matrix for Model 2 is given in Figure 10b. Moreover, we also outlined the classification report for these 2 models in Table 10 and Table 11, respectively. Results from these two models showed that, by allowing some final layers, the models updated weights according to our dataset, thereby attaining an accuracy of 93%. A comparison with existing similar systems proved the effectiveness of our proposed architecture and is highlighted in Table 12.

Our work has some limitations. Our dataset consisted of videos that contained several other frames besides the main batting shot. Hence, due to the high similarity of different batting shots (e.g., hook shot and pull shot, square cut and late cut) and having a similar direction of camera view after shots, some misclassifications were found.

## 5. Conclusions and Future Work

This paper proposed a hybrid neural network architecture that successfully classified 10 different cricket batting shots from a video, populated from publicly available data sources. We created our own batting-shot dataset consisting of various lengths, uneven illumination conditions, and overlapping scenes due to the lack of a publicly available dataset. To the best of our knowledge, the created dataset contained the 10 most unique batting shots. We aim to make our dataset public after scaling it up with more related videos to run more experiments and improve accuracy. Experiment results signified that using a pretrained model is superior when massive data are unavailable. After combining with GRU, transferring learning from a pretrained model outperformed the other models whenever the model was allowed to update its weight according to our dataset. A hybrid CNN–GRU architecture, developed from a pretrained VGG16 model, achieved the highest accuracy of 93%. We also compared our proposed architecture with similar works regarding cricket batting-shot classification, where performance results exhibited remarkable improvements to achieve state-of-the-art accuracy.

With the ongoing trend of the shorter format of cricket matches such as t20, various unorthodox shots are now being played. We plan to include these unorthodox shots in our upcoming work. Furthermore, umpire actions and boundary event classification significantly impact the generation of more accurate automatic highlight extractions. These actions are also being considered for future work. We aim to extend the dataset and build a custom model with higher accuracy. Another upcoming challenge is the detection of keyframes from a video, which allows us to work with fewer frames and parameters. Some experiments will be carried out by combining acoustic and image features. It would also be interesting to combine conventional machine-learning approaches such as SVM, scale-invariant feature transform (SIFT), and visual bag of words with deep-learning algorithms to investigate model performance. The proposed architecture can be used to solve a variety of video-classification problems. On the other hand, GPU memory is a significant impediment to the collaboration of complex models and large datasets.

## Figures and Tables

**Figure 1 sensors-21-02846-f001:**
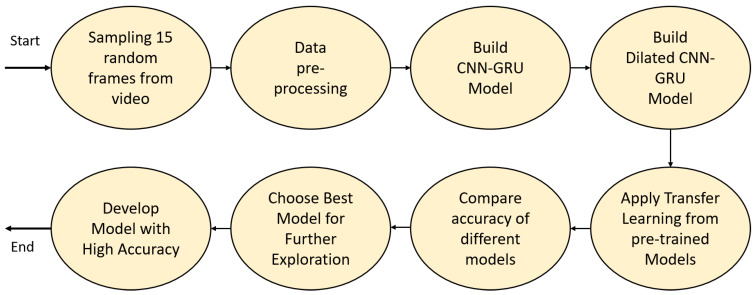
Overview of constructing the cricket batting-shot categorization model. CNN—convolutional neural network; GRU—gated recurrent unit.

**Figure 2 sensors-21-02846-f002:**
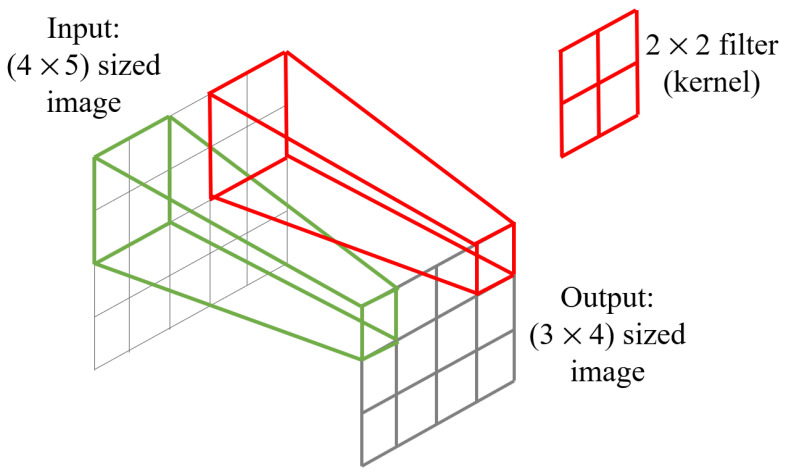
Convolutional operation using 2×2 kernel.

**Figure 3 sensors-21-02846-f003:**
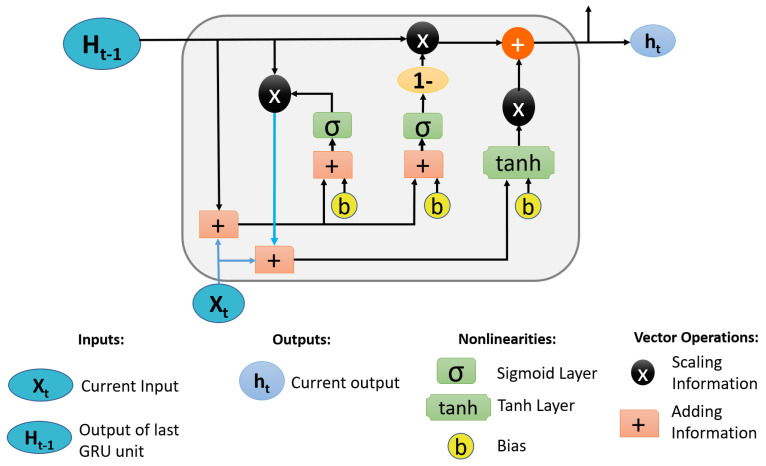
Gated recurrent unit (GRU) cell.

**Figure 4 sensors-21-02846-f004:**
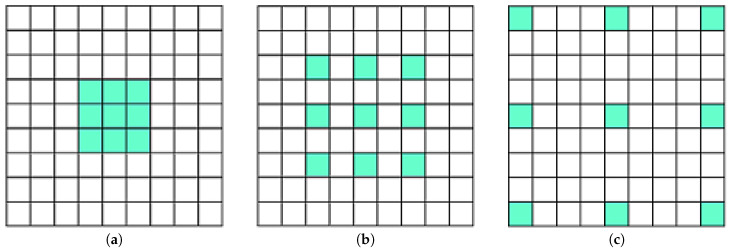
Convolution showing dilation rates of (**a**) 1, (**b**) 2, and (**c**) 4.

**Figure 5 sensors-21-02846-f005:**
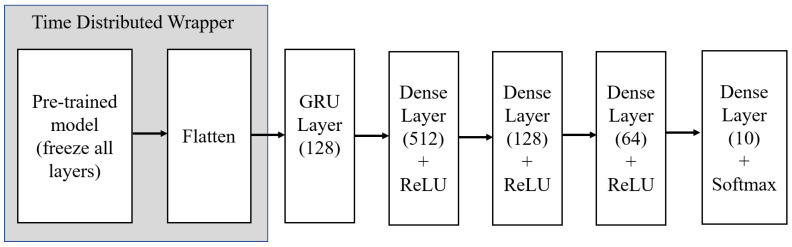
Overview of different pretrained model architectures. ReLU—rectified linear unit.

**Figure 6 sensors-21-02846-f006:**
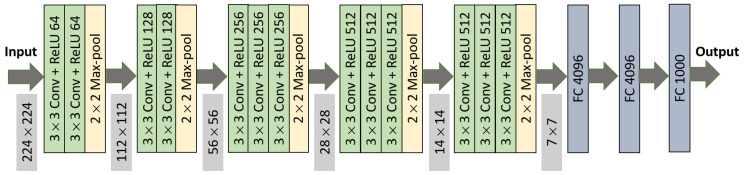
VGG16 architecture for 224×224 input image dimension.

**Figure 7 sensors-21-02846-f007:**
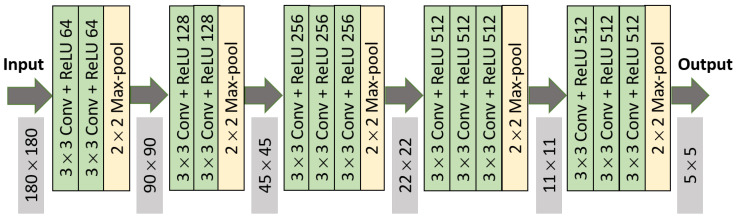
VGG16 architecture (excluding fully connected layers and 180×180 input image dimension).

**Figure 8 sensors-21-02846-f008:**
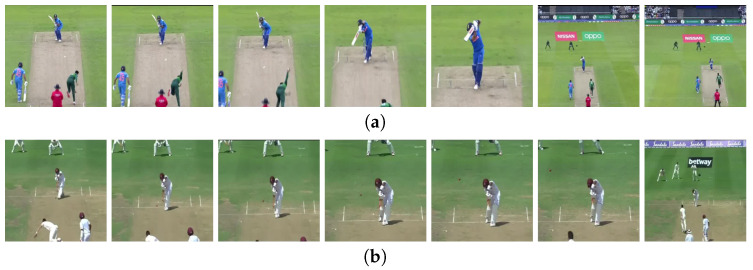
Sequential video frames for (**a**) cover drive, (**b**) defensive shot, (**c**) flick, (**d**) hook shot, (**e**) late cut, (**f**) lofted shot, (**g**) pull shot, (**h**) square cut, (**i**) straight drive, (**j**) sweep shot.

**Figure 9 sensors-21-02846-f009:**
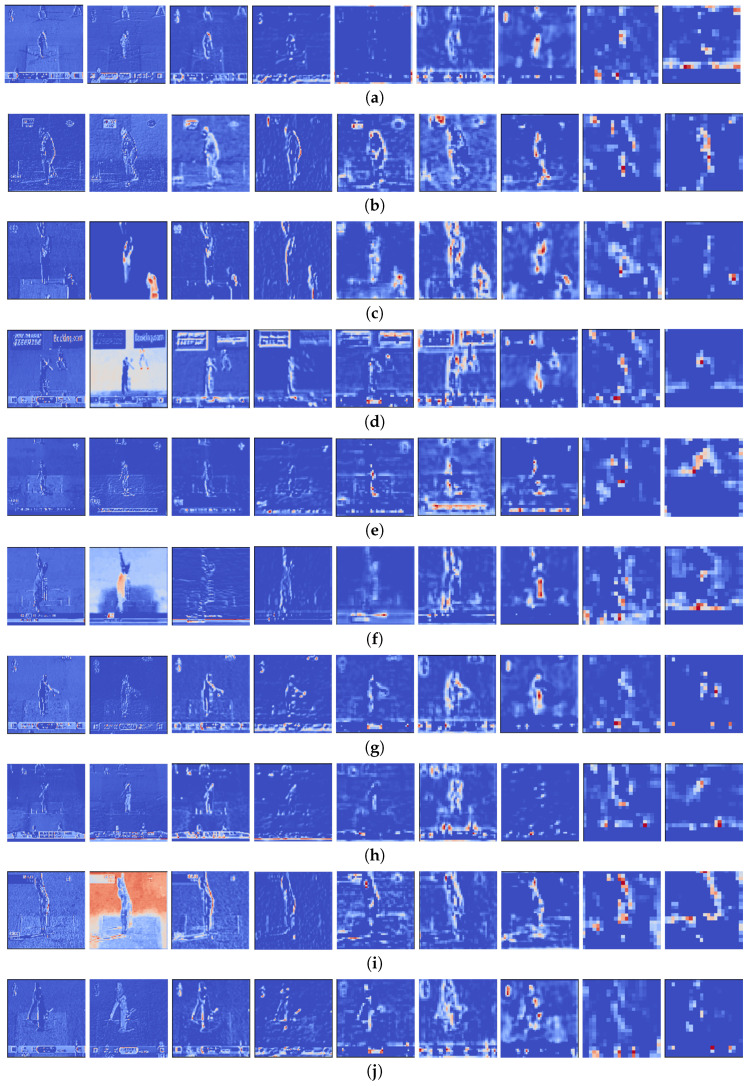
Low- to high-level feature maps using VGG16 for different batting shots: (**a**) cover drive, (**b**) defensive shot, (**c**) flick, (**d**) hook shot, (**e**) late cut, (**f**) lofted shot, (**g**) pull shot, (**h**) square cut, (**i**) straight drive, (**j**) sweep shot.

**Figure 10 sensors-21-02846-f010:**
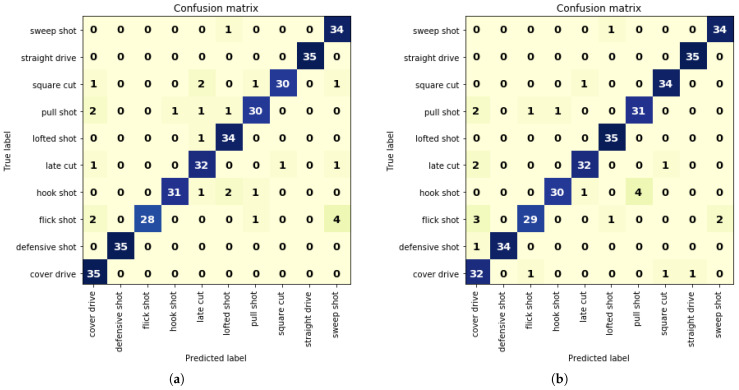
Confusion matrix for (**a**) Model 1 and (**b**) Model 2.

**Figure 11 sensors-21-02846-f011:**
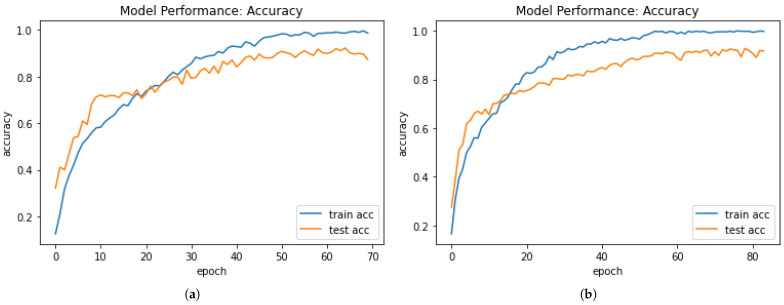
Accuracy-performance graph for (**a**) Model 1 and (**b**) Model 2.

**Table 1 sensors-21-02846-t001:** Proposed convolutional neural network (CNN) model architecture.

Layer (Type)	Output Shape	Parameters
input_1 (Input Layer)	(None, 180, 180, 3)	-
conv_1 (Conv2D)	(None, 180, 180, 64)	1792
batch_norm_1 (Batch Normalization)	(None, 180, 180, 64)	256
max_pool_1 (MaxPooling2D)	(None, 90, 90, 64)	0
conv_2 (Conv2D)	(None, 90, 90, 128)	73,856
conv_3 (Conv2D)	(None, 90, 90, 128)	1,47,584
batch_norm_2 (Batch Normalization)	(None, 90, 90, 128)	512
max_pool_2 (MaxPooling2D)	(None, 45, 45, 128)	0
conv_4 (Conv2D)	(None, 45, 45, 256)	2,95,168
conv5 (Conv2D)	(None, 45, 45, 256)	5,90,080
batch_norm_3 (Batch Normalization)	(None, 45, 45, 256)	1024
max_pool_3 (MaxPooling2D)	(None, 22, 22, 256)	0
conv_6 (Conv2D)	(None, 22, 22, 384)	8,85,120
conv_7 (Conv2D)	(None, 22, 22, 384)	13,27,488
batch_norm_4 (Batch Normalization)	(None, 22, 22, 384)	1536
max_pool_4 (MaxPooling2D)	(None, 11, 11, 384)	0
conv_8 (Conv2D)	(None, 11, 11, 480)	16,59,360
conv_9 (Conv2D)	(None, 11, 11, 480)	20,74,080
batch_norm_5 (Batch Normalization)	(None, 11, 11, 480)	1920
max_pool_5 (MaxPooling2D)	(None, 5, 5, 480)	0

**Table 2 sensors-21-02846-t002:** Model architecture for custom CNN–GRU.

Layer (Type)	Output Shape	Parameters
time_dist (custom_cnn_model)	(None, 15, 5, 5, 480)	70,59,776
time_dist (Flatten())	(None, 15, 12,000)	0
gru_1 (GRU)	(None, 180)	65,77,740
dense_1 (Dense)	(None, 512)	92,672
dense_2 (Dense)	(None, 128)	65,664
dense_3 (Dense)	(None, 64)	8256
dense_4 (Dense)	(None, 10)	650

**Table 3 sensors-21-02846-t003:** Dilated CNN (DCNN) model architecture.

Layer (Type)	Output Shape	Parameters
input_1 (Input Layer)	(None, 180, 180, 3)	-
conv_1 (Conv2D)	(None, 176, 176, 96)	2688
batch_norm_1 (Batch Normalization)	(None, 176, 176, 96)	384
max_pool_1 (MaxPooling2D)	(None, 88, 88, 96)	0
conv_2 (Conv2D)	(None, 84, 84, 128)	1,10,720
conv_3 (Conv2D)	(None, 80, 80, 128)	1,47,584
batch_norm_2 (Batch Normalization)	(None, 80, 80, 128)	512
max_pool_2 (MaxPooling2D)	(None, 40, 40, 128)	0
conv_4 (Conv2D)	(None, 36, 36, 480)	5,53,440
conv_5 (Conv2D)	(None, 32, 32, 480)	20,74,080
batch_norm_3 (Batch Normalization)	(None, 32, 32, 480)	1920
max_pool_3 (MaxPooling2D)	(None, 16, 16, 480)	0
conv_6 (Conv2D)	(None, 12, 12, 768)	33,18,528
conv_7 (Conv2D)	(None, 8, 8, 768)	53,09,184
batch_norm_4 (Batch Normalization)	(None, 8, 8, 768)	3072
max_pool_4 (MaxPooling2D)	(None, 4, 4, 768)	0

**Table 4 sensors-21-02846-t004:** Model architecture for DCNN–GRU.

Layer (Type)	Output Shape	Parameters
time_dist (dilated_cnn_model)	(None, 15, 4, 4, 768)	1,15,22,112
time_dist (Flatten())	(None, 15, 12,288)	0
gru_1 (GRU)	(None, 180)	67,33,260
dense_1 (Dense)	(None, 512)	92,672
dense_2 (Dense)	(None, 128)	65,664
dense_3 (Dense)	(None, 64)	8256
dense_4 (Dense)	(None, 10)	650

**Table 5 sensors-21-02846-t005:** VGG16–GRU-based models.

	Output From	Trainable Layers	GRU Hidden Units	Dense 1 Hidden Units	Dense 2 Hidden Units	Dense 3 Hidden Units
Model 1	block5_pool	Final 4	128	512	128	64
Model 2	block5_pool	Final 8	128	512	128	64

**Table 6 sensors-21-02846-t006:** Dataset video quantity for 10 cricket batting shots.

Name	Training Set	Test Set
Cover Drive	153	35
Defensive Shot	157	35
Flick	146	35
Hook	146	35
Late Cut	147	35
Lofted Shot	151	35
Pull	144	35
Square Cut	160	35
Straight Drive	154	35
Sweep	159	35

**Table 7 sensors-21-02846-t007:** Parameters and various accuracy measures for custom models.

Model	Trainable Parameters	Precision (%)	Recall (%)	F1 Score (%)
CNN–GRU Model	13,802,134	82.30	81.70	82.00
DCNN–GRU Model	18,419,670	83.00	82.60	83.00

**Table 8 sensors-21-02846-t008:** Parameters and various accuracy measures of pretrained models.

Model	Trainable Parameters	Precision (%)	Recall (%)	F1 Score (%)
InceptionV3–GRU	12,773,066	81.00	81.00	81.00
Xception–GRU	28,501,706	82.50	81.00	81.00
DenseNet169–GRU	16,164,554	84.90	82.90	83.00
VGG16–GRU	5,105,354	87.70	86.00	86.00

**Table 9 sensors-21-02846-t009:** Parameters and various accuracy measures of two derived models from VGG16.

Model	Trainable Parameters	Precision (%)	Recall (%)	F1-Score (%)
Model 1	12,184,778	93.00	92.60	93.00
Model 2	18,084,554	93.40	93.10	93.00

**Table 10 sensors-21-02846-t010:** Classification report for Model 1.

	Precision	Recall	F1-Score	Support
Cover Drive	0.85	1.00	0.92	35
Defensive Shot	1.00	1.00	1.00	35
Flick Shot	1.00	0.80	0.89	35
Hook Shot	0.97	0.89	0.93	35
Late Cut	0.86	0.91	0.89	35
Lofted Shot	0.89	0.97	0.93	35
Pull Shot	0.91	0.86	0.88	35
Square Cut	0.97	0.86	0.91	35
Straight Drive	1.00	1.00	1.00	35
Sweep Shot	0.85	0.97	0.91	35
Average F1-Score			0.93	350

**Table 11 sensors-21-02846-t011:** Classification report for Model 2.

	Precision	Recall	F1-Score	Support
Cover Drive	0.80	0.91	0.85	35
Defensive Shot	1.00	0.97	0.99	35
Flick Shot	0.94	0.83	0.88	35
Hook Shot	0.97	0.86	0.91	35
Late Cut	0.94	0.91	0.93	35
Lofted Shot	0.95	1.00	0.97	35
Pull Shot	0.89	0.89	0.89	35
Square Cut	0.94	0.97	0.96	35
Straight Drive	0.97	1.00	0.99	35
Sweep Shot	0.94	0.97	0.96	35
Average F1-Score			0.93	350

**Table 12 sensors-21-02846-t012:** Accuracy comparison with similar work.

Method	# of Shots Classified	Classified Shots	Accuracy (%)
Khan et al. [6]	8	Cover drive, straight drive, pull, hook, cut, sweep, onside hit, flick	90.00
Karmaker et al. [7]	4	Square cut, hook, flick, off drive	60.21
Our proposed approach	10	Cover drive, defensive shot, flick hook, late cut, lofted shot, pull, square cut, straight drive, sweep	93.00

## Data Availability

The authors have constructed a novel dataset that contains videos of 10 different cricket batting shots and termed CricShot10, which is available on request from the corresponding author.

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
