# Peer review of "CricShotClassify: An Approach to Classifying Batting Shots from Cricket Videos Using a Convolutional Neural Network and Gated Recurrent Unit"

_sensors, 2021, doi:10.3390/s21082846_

Round 1
Reviewer 1 Report
A CNN-based approach that classifies batting shots from cricket videos is proposed in this paper. The application domain seems to be interesting and it has already attracted attention by the scientific community.
I cannot recommend this paper for publication mainly because of the bad language use. I found hundreds of grammar mistakes and typos throughout the whole manuscript. The use of English is really bad and I would suggest the authors to consult a professional editor or native speaker to re-write it from scratch.
On the technical aspects, see my comments below:
- The authors proposed 2 custom CNN architectures and 4 ones relying on pre-trained CNN models. The custom CNN models actually achieved inferior performance compared with the pre-trained ones. Any reason why this happened ? The authors could comment on this in the discussion.
- What is the number of the trainable parameters in the custom CNN networks?
- I saw in the literature review that some studies used a similar approach (CNN for feature extraction followed by LSTM). How does the proposed methodology (CNN+GRU) compare with what it was presented in previous studies (e.g. CNN+LSTM)?
- The comparison between the 6 approaches (2 custom + 4 with pre-trained CNNs) was based only on the F1-scores (Table 7 and 8). The authors could also provide the rest of the performance metrics (e.g. precision, recall) for all 6 approaches as they did for models 1 and 2.
- Discussion is weak. I would like to see some critical comments with respect to the use of the proposed CNN-based approaches, explanations / reasoning why some approaches worked better than the others, why the custom CNNs failed etc.
Reviewer 2 Report
Just minor English edits are required otherwise the paper looks technical sound and just conclusions and future works can be improved.
Some recent citations need to be added.
Reviewer 3 Report
1. Section 3.3 can be improved on how to implement transfer learning models in the concerned issue and why the selected model is used, which is not clear in the paper and only Figure 5 is not enough. 2. More explanations can be given for Figure 10. Why there are two "4" values that are much higher than any others? 3. Section 5 does not hit the point of the paper and must be concise. 4. English can be improvedAuthor Response
Please see the attachment.

Round 2
Reviewer 1 Report
I would like to thank the authors for adressing my review comment. I recommend the manuscript for publication in its current status.